# Microscopic Menaces: The Impact of Mites on Human Health

**DOI:** 10.3390/ijms25073675

**Published:** 2024-03-26

**Authors:** Christina Linn, Andrea O’Malley, Kriti Khatri, Elaine M. Wright, Dylan Sebagh, Miodrag Grbić, Krzysztof Kowal, Maksymilian Chruszcz

**Affiliations:** 1Department of Biochemistry and Molecular Biology, Michigan State University, East Lansing, MI 48824, USA; plantzch@msu.edu (C.L.); omalle89@msu.edu (A.O.); khatrikr@msu.edu (K.K.); wrigh988@msu.edu (E.M.W.); sebaghdy@msu.edu (D.S.); 2Department of Biology, University of Western Ontario, London, ON N6A 5B7, Canada; mgrbic@uwo.ca; 3Department of Allergology and Internal Medicine, Medical University of Bialystok, 15-276 Bialystok, Poland; krzysztof.kowal@umb.edu.pl; 4Department of Experimental Allergology and Immunology, Medical University of Bialystok, 15-276 Bialystok, Poland

**Keywords:** Acariformes, mite, house dust mite, storage mite, chigger, *Demodex*, scabies

## Abstract

Mites are highly prevalent arthropods that infest diverse ecological niches globally. Approximately 55,000 species of mites have been identified but many more are yet to be discovered. Of the ones we do know about, most go unnoticed by humans and animals. However, there are several species from the Acariformes superorder that exert a significant impact on global human health. House dust mites are a major source of inhaled allergens, affecting 10–20% of the world’s population; storage mites also cause a significant allergy in susceptible individuals; chiggers are the sole vectors for the bacterium that causes scrub typhus; *Demodex* mites are part of the normal microfauna of humans and their pets, but under certain conditions populations grow out of control and affect the integrity of the integumentary system; and scabies mites cause one of the most common dermatological diseases worldwide. On the other hand, recent genome sequences of mites provide novel tools for mite control and the development of new biomaterial with applications in biomedicine. Despite the palpable disease burden, mites remain understudied in parasitological research. By better understanding mite biology and disease processes, researchers can identify new ways to diagnose, manage, and prevent common mite-induced afflictions. This knowledge can lead to improved clinical outcomes and reduced disease burden from these remarkably widespread yet understudied creatures.

## 1. Introduction

Mites are found in virtually every habitat across the globe: in soil and water, in the fur and skin of animals, and in the dust and stored food of human dwellings [1,2]. Mites belong to the class Arachnida, subclass Acari, and are the most diverse and abundant group of arthropods. They have been classified as hyper-diverse, with over 55,000 identified species; however, the number is likely 20 times that [3]. This lag in scientific knowledge may be due to their microscopic size [1], at only 0.5–2.0 mm in length [4].

Mite morphology consists of two main regions: the anterior prosoma, which contains the mouth parts; and the opisthosoma, which contains the legs and the eyes if present [5,6]. Mites mature through seven developmental stages which include egg, prelarva, larva, protonymph, deutonymph, tritonymph, and adult [6]. However, not all species go through every stage in a meaningful way. For some species, such as *Eutrombicula* species (spp.; chiggers), one or more of the nymph stages is passed while the exoskeleton of the previous stage is maintained [6]. Interestingly, for most mite groups, the larva has only six legs, whereas the nymph and adult have eight [3,7].

The majority of mites are free-living, not causing significant damage. Some, like the Oribatida mites, even play important ecological roles as decomposers feeding on dead plant, fungal, and animal materials [8]. Unfortunately, there are also a significant number of species that induce allergies; parasitize plants, animals, and humans; and are vectors for disease [9]. *Tetranychus urticae*, the two-spotted spider mite, is one of the most important agricultural pests as it is capable of feeding on approximately 1275 plant species, more than 150 of which are important crops [10]. *T. urticae* has been known to destroy whole fields of potatoes in Eastern Europe; they have been implicated in a 50–80% yield loss of strawberries from fields in North-Central Florida, and the estimated injury loss of tomatoes due to these mites could be as high as 50% [10]. *Ornithonyssus sylviarum*, the northern fowl mite, is the primary and most detrimental ectoparasite of commercial poultry in the United States. These parasites live their entire life cycle on their host, feeding on the host’s blood. They can cause up to 6% blood loss a day, leading to the exsanguination of heavily infested chickens, as well as inflammation and irritation [11]. Moreover, while ticks, a vector for many diseases such as Lyme disease, are included in the mite superorder Parasitiformes, they are distinct from the Acariformes that will be discussed in this review (Figure 1) [12,13].

The purpose of this article is to review mites from the Acariformes superorder that directly affect human health. Some of the major mites affecting human health around the globe are scabies mites, *Demodex* mites, chiggers, storage mites, and house dust mites (Table 1). Scabies is one of the most common dermatological diseases worldwide [16] and can cause death due to secondary infection and other complications in regions of the world where resources are scarce [16,17]. Rheumatic fever and rheumatic heart disease are two examples of such complications, and are the cause of at least 300,000 deaths every year [16]. *Demodex* mites are part of the normal commensal microfauna of human skin, but under the right conditions can grow out of control and cause ocular and dermatological disease [18]. Within the U.S., chiggers are more of a nuisance than a major health problem, but their bites can cause pruritic dermatitis and lead to secondary bacterial infections. In the Asia–Pacific region, chiggers are the sole transmitters of scrub typhus, a rickettsial-type disease that has a fatality rate of up to 50% in areas with little access to antibiotics [19,20]. Chiggers may not be a major source of health problems in the U.S., but storage mites and house dust mites are. These tiny creatures live in every home and are one of the leading causes of inhaled allergies [21]. The goal of writing this review isto highlight what is known about these major mite species and to shed light on where more research is needed for the treatment and prevention of the diseases caused by these mites.

## 2. Predominant Mite Species

### 2.1. House Dust Mites

House dust mites (HDMs) are domestic mites found ubiquitously in humid indoor areas, including beds, furniture, and carpets, and primarily feed on human or animal skin sheds [22]. Despite closely coexisting with humans, HDMs do not bite humans and, hence, are considered harmless. However, for many, HDM species are the primary source of inhalant allergies [21]. HDM feces and extracts comprise various digestive enzymes, proteases, and lipid-binding proteins that are potent allergens [23,24]. The three major species of HDM, *Dermatophagoides pteronyssinus*, *Dermatophagoides arina*, and *Euroglyphus maynei*, are major sources of inhaled allergens that affect an estimated 10–20% of the world’s population [25]. In fact, HDM allergens are the leading cause of allergic rhinitis and allergic asthma [26], which are not only major public health concerns but a huge economic burden for healthcare systems. Early life exposure to HDM allergens can lead to the development of chronic allergic asthma in adulthood [27,28]. Moreover, HDMs can be a primary sensitizer and facilitate cross-reactivity with allergens from other sources [29,30].

HDMs belong to the Pyroglyphidae family of the Astigmata order, Acari sub-class, and Arachnida class [31]. Traditionally, HDMs are not considered parasitic as they do not cause direct harm to the host (humans). However, genome sequencing suggested that HDM species evolved into free-living mites from their parasitic ancestors [32]. It is believed that HDMs developed protease-rich enzymes and characteristics such as low host specificity and humidity tolerance in order to evade their host’s immune response, characteristics that also enabled them to evolve as free-dwelling mites [33]. The parasitic lineage of HDMs is also supported by evidence that antigenic cross-reactivity has been reported between a HDM (*Dermatophagoides pteronyssinus*) and a parasitic mite, *Sarcoptes scabiei* [34]. Moreover, studies have reported that HDM allergen-specific IgE and IgG antibodies recognize antigenic targets from various helminth parasites such as lung-stage *Ascaris* larvae [35]. However, whether the antibody cross-reactivity between HDM antigens and helminth parasites has an impact on disease arising from a parasite infection is still unclear.

HDM exposure can be reduced physically by reducing humidity (<50%), proper ventilation or air-conditioning, dusting and cleaning, regularly washing bedsheets, pillowcases, and blankets, and maintaining a clean and hygienic living environment [36]. Dust mite-proof bedding options are also available, which may help minimize the dust mite population. When it comes to housing, it is suggested that avoiding the basement or first floor is better for HDM allergic patients as those areas are the most humid areas with lower air circulation [36]. HDM allergies are mostly managed using allergy immunotherapy in controlled dosages of allergen extracts, allergoids, or recombinant allergens administered to patients until they develop a clinical tolerance [37]. To avoid complications from unstandardized extracts, hypo-allergens of major HDM allergens have also been under study for clinical usage [38,39,40].

### 2.2. Storage Mites

Storage mites are very similar to HDMs, with the major difference being their primary location and some taxonomic variation. While HDMs are found almost exclusively in the home feeding on skin particles and other human debris, storage mites are found in many different grain- and food-based products such as flour, straw, and hay [41]. Storage mites can also contaminate indoor products such as dry dog food [42].

Included in the storage mite classification are two superfamilies, Glycyphagoidea and Acaroidea, with several families in each [43]. The representative storage mites with primary studies performed are *Blomia tropicalis*, *Glycyphagus domesticus*, *Lepidoglyphus destructor*, *Acarus siro*, and *Tyrophagus putrescentiae*. As with HDMs, storage mites are most commonly studied in the context of allergy, primarily causing dermatitis- or asthma-related symptoms in individuals who work with grain products or who have an infestation in household dry products [44,45,46]. To date, 44 storage mite allergens have been registered with the WHO/IUIS, with most being from *B. tropicalis* and *T. putrescentiae*.

Storage mites are sometimes implicated in other conditions aside from allergy, sometimes as a secondary symptom to the individual’s illness and usually as a non-specific invasion [43]. Storage mites have been observed in cases of pulmonary acariasis, with Acarus mites and *T. putrescentiae* sometimes implicated [47]. Storage mites, including *T. putrescentiae*, *T. longior*, *A. siro*, *Aleuroglyphus ovatus*, *Suidasia nesbitti*, *G. domesticus*, and *G. ornatus*, have been observed in cases of intestinal and urinary acariasis [48,49]. In all of these cases, antiparasitic drugs or antibiotics, such as ivermectin, chloroquine, or metronidazole, are common treatments. However, these conditions are more commonly caused by parasites such as helminths. For allergies in particular, storage mites are implicated in “pancake syndrome”, which is a host of allergic symptoms caused by reactions to mites infesting flour products with antihistamines used for the treatment of allergic conditions [50,51,52].

### 2.3. Chiggers

Trombiculid mite larvae, or chiggers, are found globally and are ectoparasites of a wide range of vertebrates. There are 50 species that bite humans, 20 of which are considered medically relevant as they cause dermatitis and/or are vectors of human pathogens [53]. The most relevant species are *Eutrombicula alfreddugesi* in North America, *Herpetecarus* spp. in South America, *Neotrombicula autumnalis* in Europe, and *Leptotrombidium* spp. in Asia [19,53]. Chiggers have four major developmental stages: egg, larva, nymph, and adult; however, only the larvae are parasitic. In the larval stage, the mite is bright red or red-brown in color and lives on leaves or grass stems close to the surface of the ground. They have six legs, are 0.25–0.5 mm long, and feed by attaching themselves to a vertebrate host as it passes through the vegetation. Chigger larvae do not feed on blood or burrow into the dermis. Instead, they pump digestive enzymes into their host, causing the partial digestion of cells at the feeding site. These liquified cells are then ingested through their stylostome, a tube formed by solidified mite saliva that extends into the dermis of the host [54]. Once fully engorged, the mite will leave the host to continue development as a predator, passing through three nymphal stages before finally molting into an adult mite [19,55]. Since only the larvae are parasitic to humans and thus vectors for bacteria and possibly viruses, we focused our attention on this developmental stage.

To date, there are no confirmed species of chiggers in the U.S. that carry zoonotic pathogens; however, this is an area that has been greatly understudied. With pathogens such as *Hantavirus*, *Bartonella*, *Borrelia*, and *Richettsia* having been detected in chiggers, there is potential for some tick-borne diseases and red-meat allergies to be caused by chiggers [56]. This does not mean they are not a nuisance in the United States, as the bites can cause pruritic dermatitis or “scrub itch”, which is caused by exposure to allergens in the mite’s saliva during feeding [19]. The treatment for chigger bites consists of topical ointments to treat itching and inflammation; in some cases, antibiotics are required for treating secondary bacterial infections [20]. Preventative measures include avoiding highly infested areas.

It is worth noting that chigger bites are sometimes confused for the bites of *Pyemotes ventricosus*, a white to yellow free-living ectoparasite that feeds on the larvae or nymphs of insects residing in seed, grain, straw, and wood. *Pyemotes ventricosus* displays very little host specificity, but if normal food sources are reduced, they may temporarily attack horses, cattle, and humans [57]. Utilizing its chelicerae, *P. ventricosus* injects a neurotoxin into the human epidermis, causing a skin eruption known as grain itch [58]. As the initial bite mimics that of chiggers, a characteristic linear serpiginous tract known as the ‘comet sign’ is pathognomonic for *Pyemotes* dermatitis. This sign may last up to 2–3 weeks after the bite and likely represents a local lymphangitis [57]. Oral antihistamines and corticosteroids have been recommended to prevent scratching and possible bacterial superinfection, and skin lesions typically self-resolve within 1–3 weeks [57,58,59,60,61].

Outside of the U.S., *Leptotrombidium* spp. found in the Asia–Pacific region are the sole vectors of *Orientia tsutsugamushi*, a Gram-negative bacterium responsible for scrub typhus [19]. In Chile, *Herpetacarus antarctica* was found to be a vector for Candidatus *O. chiloensis*, another *Orentia* bacterium that causes scrub typhus [62].

Although antigenically distinct from the Rickettsiae group, scrub typhus is a rickettsial-type disease that presents as a sudden onset of high fever, headache, lymphadenopathy, muscle soreness, rash, and a scab at the bite site. This scab at the bite site appears after a person has been bitten, but before disease onset and is thus considered an important early indicator of scrub typhus [62]. In severe cases patients also show symptoms of end-organ damage which includes hyperbilirubinemia, renal failure, and cardiovascular collapse, among others. Nearly one million cases are reported every year in the Asia–Pacific region, with farmers making up approximately two-thirds of all cases and 80% of cases occurring between the months of July and November. The mortality rate is <1% to 50% depending on antibiotic availability and treatment [63].

The most used antibiotics to treat scrub typhus are doxycycline and azithromycin, with doxycycline used for the more severe cases. The best outcomes come from treating patients for 5–7 days with either antibiotic. Alternate antibiotics for treatment include fluoroquinolones and chloramphenicol [64].

### 2.4. Demodex Mites

*Demodex* is a genus of ectoparasite mites that live in the pilosebaceous unit of mammals [65]. The major species that live on humans are *Demodex folliculorum* and *Demodex brevis*. *D. folliculorum* is 0.3–0.4 mm in length, is typically found in the upper region of the pilosebaceous unit, and is most commonly localized to the face. *D. brevis* is smaller at 0.15–0.2 mm in length, is typically found deeper in the sebaceous gland, and is most commonly found on the neck and chest [65]. *Demodex* mites are worm-like arthropods covered by a chitin exoskeleton. The mouth has chelicerae with claws and an oral needle for taking in sebum and skin cells. The opisthosoma has four pairs of legs with claws at the end [66]. *Demodex* mites have four major lifecycle stages: egg, larvae, nymph, and adult, and they live out their entire life, including mating, in the pilosebaceous unit [18].

*Demodex* species are very host specific [18,66] and considered part of the normal, healthy commensal microfauna of mammals. They generally exist in low numbers but can cause ocular or dermatological disease when they grow out of control, most often due to a compromised immune system [18]. The ocular disease caused by *Demodex* is called *Demodex* blepharitis or ophthalmic demodicosis. *Demodex* eye infestations are diagnosed through the epilation of several eyelashes which are then stained and observed under a light microscope for the presence of mites. All mites present on the lash are counted including dead mites, larvae, and eggs [67]. This method, however, can prove problematic as not all the mites are well attached to the eyelash, and many may fall off when it is removed [68]. Symptoms of ophthalmic demodicosis include irritation of the eye, burning, itching, tearing, blurry vision, and redness, swelling, or matting of the eyelid margins; another significant sign of the disease is waxy or scaly debris at the base of the eyelashes along the lid margin [69]. There are presently no clinical guidelines for the treatment of ocular demodicosis; however, various formulations of tea tree oil are most commonly used. Other treatments include terpinene-4-ol, pilocarpine 4% gel, Cilclar oxide + mercuric oxide ointment + ether, metronidazole 2% ointment, and good eyelid hygiene. Often, an antiparasitic, such as ivermectin, will be administered along with the eye ointments [68].

There are two types of dermal demodicosis: primary demodicosis, which is caused by an increase in the mite population (>5 mites/cm^2^ of skin); and secondary demodicosis, where mites inhabit the dermis; this type is thought to be the result of other skin diseases. Dermal demodicosis presents as inflammatory acne lesions of various sizes and severity, with secondary demodicosis being associated with greater inflammation and a larger distribution of the face, neck, and chest. In secondary demodicosis, mites may also introduce bacteria, such as staphylococcus, into the host [70]. Demodicosis of the skin is most frequently treated with oral or topical antiparasitic, insecticides, and antibiotics. Lotions to reduce itching are often also prescribed. These methods of treatment, however, are not always effective and can cause moderate to severe side effects [71].

### 2.5. Scabies

Human scabies is a parasitic skin infestation caused by the mite *Sarcoptes scabiei* var. *hominis*. This mite belongs to the family Sarcoptidae, subfamily Sarcoptinae. [72]. The stages of development of *S. scabiei* are consistent with other astigmatid mites: egg, larva, protonymph, tritonymph, and adult [72]. However, there are variable durations of the life cycle of *S. scabiei* var. *hominis*. The observed variations can likely be ascribed to the challenges associated with in vivo observation of *S. scabiei* var. *hominis* in the skin. Other factors contributing to the observed differences in this life cycle could be inconsistent observation methods, the temperature and humidity conditions during observation, and the length of observational periods [72]. Scabies mites penetrate and burrow into the stratum corneum of the epidermis of mammalian skin. Once burrowed, the mites ingest and feed on intracellular fluid from the broken cells that fill the burrow. The mites will continue to burrow deeper towards the dermis as the upper layers of the skin are pushed towards the surface [72]. 

Scabies is one of the most common dermatological diseases worldwide and is most prevalent in tropical communities with limited access to resources and healthcare services [16]. The disease predominately affects children, elderly populations, and immunocompromised individuals, and is transmitted through prolonged close skin contact or sexual contact [73]. The diagnosis of scabies is often difficult due to delayed and varied symptom presentation. Upon primary infestation, most individuals are asymptomatic and can be infected up to eight weeks before symptoms present, when a diagnosis is made. Common scabies symptoms include a severe itchy rash that worsens at night, and inflamed regions and papules on the hands, feet, ankles, genitalia, and scalp [16,74]. In cases of reinfestation, symptoms can present as early as one or two days after and may be worsened by the host’s hypersensitivity [16]. A scabies diagnosis is confirmed by identifying the scabies mite, its eggs, or fecal pellets (referred to as “scybala”) through microscopic analysis. However, as this confirmation may not be easily obtained due to the limited presence of mites in individuals with classic scabies, and because a microscopic examination may not always be available, a provisional diagnosis is occasionally made by relying on a consistent medical history and physical examination [75].

Immunocompromised individuals are more likely to develop a more severe form of scabies, known as crusted or Norwegian scabies [74]. Crusted scabies infestations for this community are considered life threatening and often lack the usual symptoms of scabies, such as the characteristic rash and itching. Instead, symptoms for these individuals present as thick layers of crusted skin that flake off and host large numbers of scabies mites and eggs. These patients can host up to millions of mites in the skin during infestation compared to the lower mite burden observed in classic scabies [75]. This primary change in symptom presentation results in the disease being more easily transmitted, requiring less skin-to-skin contact and through handling of contaminated items such as bedding and clothing [75]. Crusted scabies often leads to complications resulting in secondary bacterial infections, such as impetigo, which is caused by *Streptococcus pyogenes*. Impetigo from *S. pyogenes* is often a precursor to multiple other infections, diseases, and autoimmune complications such as invasive *S. pyogenes*, glomerulonephritis, rheumatic fever, and recurrent complicated skin infections [16].

Treatments for scabies infestation are only available as prescribed products and medications. Scabicides, products that kill scabies mites and mite eggs, are lotions or creams to be applied to affected regions [76]. Some of the most common lotions and creams are permethrin cream (5%), crotamiton cream (10%), benzyl benzoate lotion (25%), spinosad liquid (0.9%), precipitated sulfur cream (6–33%), and lindane lotion (1%), all of which are prescribed depending on the age range of the patient and the patient’s reaction to topical treatment. In more severe cases of scabies infestations, scabicides alone may not be enough. In this case, a repeat treatment with scabicides and an oral antiparasitic, such as ivermectin, can be used. Additionally, individuals with crusted scabies may be prescribed antibiotics for any secondary infections caused by the disease [74,75,76].

Treatment for scabies is recommended not only for the infected individual, but also for all persons who have had close contact with the infected individual, such as household members, close personal contacts, and sexual partners. The treatment of all individuals at the same time is strongly encouraged to prevent reinfestation. To further prevent reinfestation, it is suggested that all bedding, clothing, and towels used by infested individuals be decontaminated by washing and drying linens or clothes in hot conditions [76].

## 3. Studies of Mites at the Molecular Level

As already mentioned, despite their abundance in the environment and significant impact on humans, animals, and plants, mites are understudied. This is especially true when we consider the molecular basis of parasitic mites’ lives and their interactions with their hosts. Currently, the NCBI contains 116 genomes for mites in the Acariformes superfamily, with 19 of these being from the Trombidiformes order and 97 from the Sarcoptiformes. Overall, these genomes represent 14 Trombidiformes mites and 77 Sarcoptiformes mites, as several mites have had their genomes sequenced several times. Of these mites, seven are implicated in allergy (*T. putrescentiae*, *D. farinae*, *D. pteronyssinus*, *B. tropicalis*, *E. maynei*, *L. destructor*, and *T. urticae*), two are vectors for scrub typhus (*Leptotrombidium delicense* and *L. pallidum*), and one is responsible for scabies (*S. scabiei*) [77]. While there are no officially registered allergens originating from *T. urticae*, proteins originating from this agricultural pest can on rare occasions cause sensitization in farmers, orchard workers, and green house workers [78,79,80,81,82]. An evaluation of the available mite genomes shows that, of the approximately 55,000 known species, less than 0.2% have had their genomes sequenced and only 10 are Acarifomes known to directly impact human health. Interestingly, of the 116 genomes available, 97 were published after 2020, indicating that the molecular study of mites is growing.

Proteins originating from mites are also understudied in terms of structural biology. For example, the Protein Data Bank (PDB) exhibits a notable disparity in the representation of mite-derived proteins [83], with a mere 57 structures. Of these, 43 structures are for proteins classified in the Sarcoptiformes order, including 17 structures from *D. pteronyssinus*, 12 structures from *D. farinae*, 9 from *B. tropicalis*, 3 from *T. putrescentiae*, and 2 from *S. scabiei*. Of these, seven are allergen–antibody complexes; six are with murine IgG (binding Der p 1, Der p 2, or Der f 1), and one is with human IgE (in complex with Der p 2) [38,84,85,86]. In fact, every protein from Sarcoptiformes represented in the PDB was studied in the context of an HDM or storage mite allergy, with the two proteins from *S. scabiei* homologous to house dust mite allergens. The PDB contains 12 structures of proteins originating from *T. urticae*, which is the only representative of the Trombidiformes order. Finally, two structures in the PDB are of odorant-binding proteins from *Varroa destructor*, which is included in the Mesostigmata order. Figure 2 shows the structures of the representative mite proteins.

With much work to be performed in the study of mites, and with the sequence of the first chelicerate and mite genome, that of *T. urticae* [87], we can envisage the impact of genomics in this field. This includes the development of new methods based on RNAi to control mites [88] that may function across species. Furthermore, the development of new materials derived from mites, such as spider mite silk and nanoparticles developed from the silk [89], will expand the utilization of mite-derived materials in biomedicine. Thus, research on mites is entering a dynamic period where these organisms with sequenced genomes will provide novel tools for their control as well as the application of mite-derived products in different, non-traditional fields.

## 4. Conclusions

Mites belonging to the class Arachnida, subclass Acari, are a hyper-diverse group with a global presence. Currently, there are approximately 55,000 described species, but it is believed that hundreds of thousands more species are yet to be discovered [2,3]. While most mites are harmless and serve ecological roles, some species are detrimental, causing allergies; parasitizing plants, animals, and humans; and acting as vectors for diseases. Five of the major mites implicated in global disease are house dust mites, storage mites, chiggers, *demodex* mites, and scabies mites. Although these, and possibly other mites, are of high medical importance, research is greatly lacking when compared to that of other arachnids [5]. One area that is particularly lacking in Acarology is the structural and functional characterization of mite proteins. Most of the work performed in this area is related to mites that induce allergies, but little has been carried out for mites that are ectoparasites of humans and animals. Proteins are fundamental macromolecules playing crucial roles in immunity, catalysis, metabolism, and the majority of other biological processes [90]. Furthermore, because the majority of physiological and disease processes are manifested within/through proteins, understanding their structure and function is crucial to disease prevention and treatment.

## Figures and Tables

**Figure 1 ijms-25-03675-f001:**
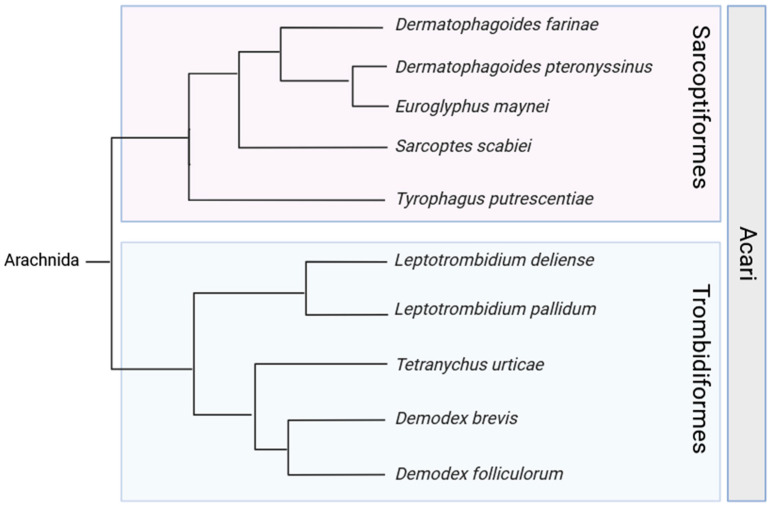
A diagram showing the relationship between the major mite species mentioned in this review [14,15] (created in BioRender).

**Figure 2 ijms-25-03675-f002:**
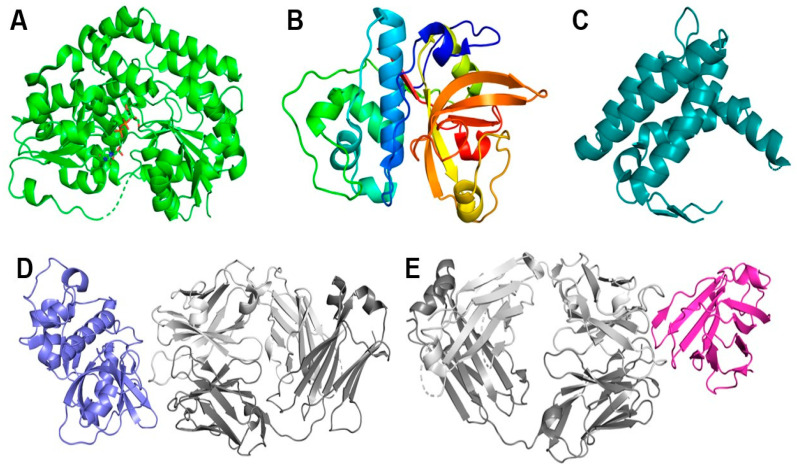
Structure of proteins found in major mite species. (**A**) UDP-glycosyltransferase TuUGT202A2 from *Tetranychus urticae* binding UDP-glucose (PDB: 8SFY); (**B**) major house dust mite (*Dermatophagoides pteronyssinus*) allergen Der p 1 (PDB: 2AS8); (**C**) odorant-binding protein 1 from *Varroa destructor* (PDB: 7NYJ); (**D**) major house dust mite (*D. farinae*) allergen Der f 1 (slate blue) in complex with murine-derived IgG Fab (light chain in dark grey, heavy chain in light grey) (PDB: 5VPL); (**E**) major house dust mite (*D. pteronyssinus*) allergen Der p 2 (magenta) in complex with specific human IgE Fab (light chain in dark grey, heavy chain in light grey) (PDB: 7MLH).

**Table 1 ijms-25-03675-t001:** Table of major mite species affecting humans and the diseases associated with them.

Class	Family	Species	Disease
Arachnida	Trombiculidae	*Leptotrombidium delicense*	Scrub typhus
		*Leptotrombidium pallidum*	
		*Leptotrombidium akamushi*	
		*Leptotrombidium rubellum*	
Arachnida	Demodicidae	*Demodex brevis*	Demodicosis
		*Demodex folliculorum*	
Arachnida	Sarcoptidae	*Sarcoptes scabiei*	Scabies
Arachnida	Pyroglyphidae	*Dermatophagoides pteronyssinus*	Allergy
		*Dermatophagoides farina*	
		*Euroglyphus maynei*	
Arachnida	Glycyphagoidea	*Blomia tropicalis*	Allergy
		*Glycyphagus domesticus*	
		*Lepidoglyphus destructor*	
Arachnida	Acaroidea	*Acarus siro*	Allergy
		*Tyrophagus putrescentiae*	

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
