# Peer review of "Microscopic Menaces: The Impact of Mites on Human Health"

_ijms, 2024, doi:10.3390/ijms25073675_

Round 1

Reviewer 1 Report

Comments and Suggestions for Authors

Dear Authors, 

My main comments are:

- You should better focus on the topic of the paper, i.e., mite groups that are with impact on human health.  You can mention other groups in the introduction, but not in later sections (unless there would be a connection to the topic) .

- I was very surprised you did not include ticks in your review. If you did it on purpose and there is an important reason for that - it should be mentioned at the beginning. 

- I was quite dissapointed with the section: Studies of Mites at the Molecular Level. But maybe the state of knowledge is really so poor and it is not your fault... You mention only 'The Protein Data Bank', what with GenBank? Are there any sequencies of mite groups you write about? If yes, I suggest to give a thorough overview, maybe in table format? 

Other comments are included in the text enclosed. 

Author Response

My main comments are:

You should better focus on the topic of the paper, i.e., mite groups that are with impact on human health.  You can mention other groups in the introduction, but not in later sections (unless there would be a connection to the topic) .

Thank you for noticing it. We have included additional information that is for example related to T. urticae and allergy among farmers, orchad and green houses workers.

I was very surprised you did not include ticks in your review. If you did it on purpose and there is an important reason for that - it should be mentioned at the beginning. 

Thank you for pointing it out. In fact we have omitted ticks on purpose. This review focuses on species from the Acariformes superorder. We have clarified this in the text of the manuscript.

I was quite dissapointed with the section: Studies of Mites at the Molecular Level. But maybe the state of knowledge is really so poor and it is not your fault... You mention only 'The Protein Data Bank', what with GenBank? Are there any sequencies of mite groups you write about? If yes, I suggest to give a thorough overview, maybe in table format? 

Thank you for this comment. We have added information on mite genomes. Generally, mites are very poorly studied on a molecular level. It seems that only T. urticae has a bigger scientific community working on this organism at the molecular level.

Reviewer 2 Report

Comments and Suggestions for Authors

In the reviewed MS the authors highlight what is known about mites that directly affect human health they also aimed “to bring to light where more research is needed to help not only treat the diseases caused by these mites but to help prevent them”

The MS is clearly written. Stylistically it is a semi-popular review. It is quite detailed in mite biology but very poor in molecular/genomic aspects, that are the main scope of the journal.

The text (beside the poorly written section 3) needs some minor revisions.

The section 3 should be expanded and more comprehensive review on “molecular aspects” should be given.

Finally, this MS would notably benefit from bright figures, showing the mites and illustrating the key findings/directions in molecular research on mites (poorly covered in the current version of the MS)

34 – lots of mites are associated with plants and there are a many phytophagous pests, this may be mentioned here

38-39 – some cautions should be kept in mind considering very high number of described and undescribed spiders, which are also arachnids

41 There are mite groups for which gnathosoma/idiosoma is inapplicable. Probably prosoma/opisthosoma is more universal.

43 – 7 developmental stages, please give a reference

45 – Eutrombicula – italic

47-48 – did you review 6/8 leg problem in mesostigmatid mite larvi?

49-50 – oribatids may be associated with parasitic worms dangerous to humans

68 – Demodex italic

98 – please give more precise taxonomy for Phyroglyphidae, Arachnida is too distant

101 – 26 is the correct citation here, not 25

108 Ascaris italic

111-116 – please add supportive references

139 Acarus italic

141 – each Genus/species should be accompanied with Author and Year (check through the whole text)

142-143 please provide a reference

158-160 – this is the place where a reader would like to see some images. There a several other similar places. Please, consider inserting bright illustration, they will definitely make this review more attractive

163 – please, say briefly what stylostome is, otherwise unprepared reader (your main audience according to the style of the MS) may be confused.

169- Hantavirus, Bartonella, Borrelia, and Richettsia – italic

176 give an image/microphotograph

178  - start a new sentence with full word, not abbreviation (P.)

180 – neurotoxin of unknown nature? Can you say some words about this neurotoxin?

184 Pyemotes italic

192 Orentia italic

209, 215 ….. Demodex italic (check everywhere)

215-217 these sentences need revision

314 this section is very short, especially considering the scope of the journal

Comments on the Quality of English Language

no comments

Author Response

Thank you very much for all comments. We hope that majority of them were addressed, and the new version is significantly improved.

34 – lots of mites are associated with plants and there are a many phytophagous pests, this may be mentioned here

To avoid potential confusion, we have listed only T. urticae which is also implicated in some allergic reactions among farmers, orchard and green houses

38-39 – some cautions should be kept in mind considering very high number of described and undescribed spiders, which are also arachnids

Thank you for this comment. It was clarified.

41 There are mite groups for which gnathosoma/idiosoma is inapplicable. Probably prosoma/opisthosoma is more universal.

Thank you for this comment. We have updated the terminology.

43 – 7 developmental stages, please give a reference

Done

45 – Eutrombicula – italic

Done

47-48 – did you review 6/8 leg problem in mesostigmatid mite larvi?

Clarified

49-50 – oribatids may be associated with parasitic worms dangerous to humans

Thank you for bringing it up. However, due to the focus of the manuscript we

have not mentioned it.

68 – Demodex italic

Thank you for noticing it. It was corrected.

98 – please give more precise taxonomy for Phyroglyphidae, Arachnida is too distant

Done

101 – 26 is the correct citation here, not 25

Corrected

108 Ascaris italic

Done

111-116 – please add supportive references

Done

139 Acarus italic

Done

141 – each Genus/species should be accompanied with Author and Year (check through the whole text)

142-143 please provide a reference

Done

158-160 – this is the place where a reader would like to see some images. There a several other similar places. Please, consider inserting bright illustration, they will definitely make this review more attractive

Unfortunately we do not have such illustrations.

163 – please, say briefly what stylostome is, otherwise unprepared reader (your main audience according to the style of the MS) may be confused.

Done

169- Hantavirus, Bartonella, Borrelia, and Richettsia – italic

Done

176 give an image/microphotograph

Unfortunately, we do not have such an image

178 - start a new sentence with full word, not abbreviation (P.)

Done

180 – neurotoxin of unknown nature? Can you say some words about this neurotoxin?

Unfortnately, we were not able to find any information on the neurotoxin.

184 Pyemotes italic

Done

192 Orentia italic

Done

209, 215 ….. Demodex italic (check everywhere)

Thank you for noticing it. It was corrected.

215-217 these sentences need revision

Corrected

314 this section is very short, especially considering the scope of the journal

The section was extended

Round 2

Reviewer 2 Report

Comments and Suggestions for Authors

48,49 - this is incorrect sentence. Please revise it according to any classical manual of Acarology, e.g. Krantz, Walter 2009

Figure1 - this figure need revision, please give it in accordance with recent phylogenetic data on Acari and Chelicerata, otherwise it looks wierd

333 - why in the review of 2024 you give information fro 2021? Please, give the most recent information on the number of the genomes here, not old information.

360 "Though we are a long way from where we would like to be" --- and some other similar places in the text: please check if this is stylistically appropriate, some parts of the text sound like a newspaper article.

Comments on the Quality of English Language

"Though we are a long way from where we would like to be" --- and some other similar places in the text: please check if this is stylistically appropriate, some parts of the text sound like a newspaper article.

Author Response

48,49 - this is incorrect sentence. Please revise it according to any classical manual of Acarology, e.g. Krantz, Walter 2009

Corrected

Figure1 - this figure need revision, please give it in accordance with recent phylogenetic data on Acari and Chelicerata, otherwise it looks wierd

Figure was updated

333 - why in the review of 2024 you give information fro 2021? Please, give the most recent information on the number of the genomes here, not old information.

Thank you for this comment. We have updated this section with current information that is available in GenBank.

360 "Though we are a long way from where we would like to be" --- and some other similar places in the text: please check if this is stylistically appropriate, some parts of the text sound like a newspaper article.

We have further reviewed the paper and made some adjustment. I hope that it reads better now.